Antifungal activity and mechanism of novel peptide Glycine max antimicrobial peptide (GmAMP) against fluconazole-resistant Candida tropicalis

Cai Ruxia 1
Zhao Na 1 2
Sun Chaoqin 1
Huang Mingjiao 1 3
Jiao Zhenlong 1 3
Peng Jian 1 2
Zhang Jin 4
Guo Guo 1 2 guoguojsc@163.com
1 School of Basic Medical Sciences, Guizhou Key Laboratory of Microbial and Infectious Disease Prevention & Control, Guizhou Medical University , Guiyang, Guizhou , China
2 Key Laboratory of Environmental Pollution Monitoring and Disease Control (Guizhou Medical University), Ministry of Education , Guiyang, Guizhou , China
3 Translational Medicine Research Center, Guizhou Medical University , Guiyang, Guizhou , China
4 School of Public Health, Guizhou Medical University , Guiyang, Guizhou , China
Keller Nancy
Electronic publication date: 2025 May 20
Publication date: 2025
Volume: 13
Electronic Location ID: e19372
Received 2024 Nov 18; Accepted 2025 Apr 4
Copyright: © 2025 Cai et al.
Copyright year: 2025
Copyright holder: Cai et al.
License: This is an open access article distributed under the terms of the Creative Commons Attribution License, which permits unrestricted use, distribution, reproduction and adaptation in any medium and for any purpose provided that it is properly attributed. For attribution, the original author(s), title, publication source (PeerJ) and either DOI or URL of the article must be cited.
License URL: https://creativecommons.org/licenses/by/4.0/

Keywords: Antimicrobial peptide, GmAMP, Drug-resistance, Antifungal activity, Candida tropicalis

Funding: National Natural Science Foundation of China 81760647 and 82360700 Science and Technology Planning Project of Guizhou Province ZK[2022]345 Excellent Young Talents Plan of Guizhou Medical University [2021]104 Guizhou Key Laboratory ZDSYS[2023]004 This research received funding from the National Natural Science Foundation of China (No. 81760647, 82360700), Science and Technology Planning Project of Guizhou Province (ZK[2022] general project 345), Excellent Young Talents Plan of Guizhou Medical University (No. [2021]104) and Guizhou Key Laboratory (ZDSYS[2023]004). The funders had no role in study design, data collection and analysis, decision to publish, or preparation of the manuscript.

==============================
Background

There is a pressing need to create innovative alternative treatment approaches considering the overuse of antifungal drugs causes the number of clinically isolated fluconazole-resistant Candida species to increase. Glycine max antimicrobial peptide (GmAMP) is a novel peptide screened by us using artificial intelligence modeling techniques, and pre-tests showed its strong antimicrobial activity against clinically fluconazole-resistant Candida tropicalis.

Methods

The study aimed to comprehensively investigate the antimicrobial activity and mechanisms of GmAMP against fluconazole-resistant C. tropicalis. The antifungal activity of GmAMP against fluconazole-resistant C. tropicalis was assessed by using broth microdilution method, growth and fungicidal kinetics, hypha transformation, and antibiofilm assay. To further uncover the potential mechanisms of action of GmAMP, we performed scanning electron microscopy, flow cytometry, cell membrane potential probe 3, 3′-Dipropylthiadicarbocyanine Iodide (DiSC3(5)), and reactive oxygen species (ROS) probe 2′, 7′-Dichlorodihydrofluorescein diacetate (DCFH-DA) detection to assess the cellular morphology and structure, membrane permeability, membrane depolarization, and ROS accumulation, respectively. At the same time, we used cytotoxicity and degree of erythrocyte hemolysis assays to assess GmAMP’s toxicity in vitro. Cytotoxicity and treatment efficacy were evaluated in vivo by utilizing the Galleria mellonella larvae infection model.

Results

GmAMP exhibited significant antifungal activity against fluconazole-resistant C. tropicalis with a minimum inhibitory concentration (MIC) of 25 µM and demonstrated fungicidal effects at 100 µM within 2 h. GmAMP prevented the transition from yeast to hypha morphology, inhibited the biofilm formation rate of 88.32%, and eradicated the mature biofilm rate of 58.28%. Additionally, GmAMP treatment at 100 µM caused cell structure damage in fluconazole-resistant C. tropicalis, whereas GmAMP treatment at concentrations ranging from 25 to 100 µM caused membrane permeability, depolarization of cell membrane potential, and intracellular ROS accumulation. Moreover, GmAMP enhanced the survival rate of 75% for G. mellonella with fluconazole-resistant C. tropicalis infection as well as reduced fungal burden in vivo by approximately 1.0 × 102 colony forming units per larva (CFU per larva).

Conclusion

GmAMP can disrupt the cell membrane of fluconazole-resistant C. tropicalis and also shows favorable safety and therapeutic efficacy in vivo. Accordingly, GmAMP has the potential to be an agent against drug-resistant fungi.

Introduction

Opportunistic fungal pathogens can cause cutaneous infections and invasive infections, posing a significantly higher risk of morbidity and mortality in immunocompromised patients (Pathakumari, Liang & Liu, 2020). Among these pathogens, invasive candidiasis emerges as a prominent concern, characterized by mortality rates of approximately 25% (Pfaller & Diekema, 2007; Sasani et al., 2021). Despite being a component of human microbiomes, Candida tropicalis is considered to be the second to fourth most virulent genus of Candida species and may result in invasive infections (Zuza-Alves, Silva-Rocha & Chaves, 2017). Azole antifungal agents, including fluconazole, itraconazole, voriconazole, posaconazole, and others, constitute the primary therapeutic resources against C. tropicalis infection. However, the indiscriminate and excessive use of azoles has engendered the emergence of azole-resistant strains of C. tropicalis from clinical isolates. In addition, compared to other Candida species, C. tropicalis showed a greater rate of fluconazole resistance, according to epidemiologic research (Tseng et al., 2022). Consequently, the significant prevalence of azole resistance in clinical isolates of C. tropicalis has drawn more attention to invasive infection (Fan et al., 2023). C. tropicalis was included as a high-priority pathogen in the World Health Organization’s (WHO) first list of fungal priority pathogens in 2022, which was intended to create a strategic framework for research, development, and public health interventions (Fisher & Denning, 2023; World Health Organization (WHO) ARDA, Control of Neglected Tropical Diseases (NTD) & Global Coordination and Partnership (GCP), 2022).

It is widely recognized that azoles, fluconazole in particular, are crucial are essential preventive and therapeutic medicines for the management of Candida infections. Despite the widespread and regular use of fluconazole, resistance is increasing, which has caused the emergence of cross-resistance to other antifungal medications (Forastiero et al., 2013). Studies have shown that resistance rates of C. tropicalis to azoles such as fluconazole, itraconazole, voriconazole, and posaconazole have recorded resistance rates as high as 40% to 80% (World Health Organization (WHO) ARDA, Control of Neglected Tropical Diseases (NTD) & Global Coordination and Partnership (GCP), 2022). Furthermore, more than 21% of C. tropicalis isolates in China are resistant to fluconazole and even 21.7% of C. tropicalis isolates were resistant to 2–4 azoles in Iran (Badiee et al., 2022; Liu et al., 2022). A number of mechanisms, such as changes in drug targets, elevation of drug target expression, and increased expression of efflux pumps, may be responsible for the resistance phenomena (Lee et al., 2021). As a result, antifungal drug therapies may be inefficient in treating drug-resistant candidiasis, which poses significant challenges in clinical management (McCarthy & Walsh, 2017; Zhang et al., 2017). In order to address the current issue of fungal drug resistance, it is imperative to utilize both novel and promising antifungal medicines as well as alternative treatment techniques.

Antimicrobial peptides (AMPs) represent critical components of the immune defense system in organisms, with broad-spectrum antimicrobial activity, low drug resistance, and diverse bactericidal mechanisms (Li et al., 2020). Antimicrobial peptides can destroy multiple targets of pathogens through membrane and non-membrane interactions, which differs from the single target bactericidal principle of conventional antibiotics. The membrane destructive effect is primarily exerted through three mechanisms: barrel-stave, toroidal or carpet, these unique mechanism makes it not easily susceptible to the development of resistance (Gan et al., 2021). We have previously reported that we employed multitasking adaptive modeling and model adaptation to establish a prediction model and screening protocol for antifungal peptides based on antimicrobial peptide databases such as APD, DRAMP, CAMP, antifp, etc. (Zhang et al., 2022). Then we predicted more than three million unknown functional sequences in the UniProt database from the established model and screened out several hundreds of peptides that might be antifungally active then synthesized them by solid-phase organic synthesis method, and the antimicrobial activity was verified by wet lab experiments. The novel antimicrobial peptide SPGKKKKKKKKKKKTKKKKKK showed strong antimicrobial activity against fluconazole-resistant C. tropicalis in the pre-test, with a minimum inhibitory concentration (MIC) of 25 µM (MIC of fluconazole against this isolate >3,343 µM).

According to the sequence homology nomenclature in the antimicrobial peptide APD3 database (http://aps.unmc.edu/AP/), the Glycine max antimicrobial novel peptide was named GmAMP. In this report, to more fully assess GmAMP, we first determined the MIC of GmAMP against four strains of clinically fluconazole-resistant C. tropicalis and the standard strain of C. tropicalis ATCC 20962, for which GmAMP showed excellent antimicrobial activity. Besides, to assess the antifungal activity and mechanism of GmAMP, we performed experimental validation on fluconazole-resistant C. tropicalis. As a result, the antifungal mechanism of GmAMP against fluconazole-resistant C. tropicalis was investigated in terms of physicochemistry and morphology, and its in vivo efficacy was evaluated by the G. mellonella larvae infection model. Thus, the findings of this study provide a certain experimental foundation for the exploration and utilization of novel peptide antimicrobial drugs.

Materials and Methods

Materials

The peptide GmAMP (SPGKKKKKKKKKKKTKKKKKK) was synthesized by solid phase chemical synthesis method by Gil Biochemical Co., Ltd (Shanghai, China), purified by reversed-phase high-performance liquid chromatography (RP-HPLC) (Fig. S1A), and the purity was >95%. They were dissolved in deionized water at a stock concentration of 5 mg/mL before use. The molecular weight of GmAMP was determined using electrospray ionization (ESI) mass spectrometer (ESI MS) (Fig. S1B). We used the online website Heliquest software (https://heliquest.ipmc.cnrs.fr/index.html) to predict the helix diagram and hydrophilicity, and AlphFold2 to predict the 3D structure of peptides. The molecular weight of the peptide was predicted by Expasy ProtParam (https://web.expasy.org/protparam/).

Strains and cell culture conditions

The fluconazole-resistant C. tropicalis 4171, 4252, 6984, and 8402 were collected from infected patient’s blood in the affiliated hospital of Guizhou Medical University. C. tropicalis ATCC 20962 was purchased from the Shanghai Conservation Biotechnology Center. All strains were grown in yeast extract peptone dextrose medium (YPD, Solarbio, Beijing, China) at 35 °C with shaking at 200 rpm until the cultures reached the logarithmic phase. RPMI-1640 (Invitrogen, Carlsbad, CA, USA) supplemented with 15% fetal bovine serum (FBS) (Sigma-Aldrich) was used as the culture medium for hypha growth of C. tropicalis. The mouse monocyte-macrophage cell line RAW 264.7 was donated by Jiahong Wu from the Key and Characteristic Laboratory of Modern Pathogen Biology, Guizhou Medical University, and cells were cultured in DMEM medium (Gibco, Waltham, MA, USA) containing 10% fetal bovine serum (FBS, Gibco), 100 U/mL penicillin (Gibco, Waltham, MA, USA), and 100 µg/mL streptomycin (Gibco, Waltham, MA, USA) and maintained at 37 °C in a humidified 5% CO2 incubator.

Antifungal activity

The minimum inhibitory concentration (MIC) of GmAMP for four fluconazole-resistant C. tropicalis from clinical isolates, and a standard strain of C. tropicalis ATCC 20962 was determined by using broth microdilution method according to the Standards of Clinical and Laboratory Standards Institute (CLSI) (Clinical and Laboratory Standards Institute (CLSI), 2023). In brief, these yeasts of fungal strains were cultured in YPD broth medium at 35 °C to the logarithmic growth stage, and the cultures were washed by phosphate-buffered saline (PBS, 10 mM, pH 7.4) three times and resuspended to 0.5 × 103~2.5 × 103 CFU/mL, then 100 µL above fungal suspension was added to a 96-well plate with a series concentration of GmAMP (100, 50, 25, 12, 6, 3 µM) or fluconazole (3,343 µM to 3 µM) (Yuan Ye, Shanghai, China) and 10 mM PBS and medium were used as the negative and blank controls, respectively. After co-incubation at 35 °C for 24 h, The drug concentration corresponding to the well without visible fungal growth was regarded as MIC, corresponding to 90% inhibition of fungal growth. The experiment was performed in triplicate and repeated three times.

Growth and fungicidal kinetics

To analyze the antifungal or fungicidal process of GmAMP against fluconazole-resistant C. tropicalis, the growth kinetics and time-kill kinetics of GmAMP on fluconazole-resistant C. tropicalis were further investigated at different times after GmAMP treatment as to previously described (Ramesh et al., 2023). The concentration of the prepared fungal cells was 1.0 × 106 CFU/mL according to the previously mentioned method and incubated with 25, 50, and 100 µM of GmAMP at 35 °C for 48 h. A microplate reader (Thermo Fisher Scientific, Waltham, MA, USA) was used to record the OD630nm every 2 h during co-cultivation. In the meantime, the cultivated yeasts were harvested at certain intervals (0, 2, 4, 6, 8, 10, and 12 h), and the agar solid plate experiment was conducted following gradient dilution. The negative and blank controls in the experiment were 10 mM PBS and medium. Fungal colonies were counted after incubation at 35 °C for 24 h. The results were presented as the average of triplicate measurements from three independent assays.

Effect of GmAMP on hypha formation

To analyze the effect of GmAMP on the transition of yeast-to-hyphal phase in fluconazole-resistant C. tropicalis as described previously (Jiang et al., 2016). The concentration of the prepared fungal cells was 1.0 × 106 CFU/mL in RPMI 1640 medium (Gibco, Waltham, MA, USA) which contained 15% fetal bovine serum (Gibco, Waltham, MA, USA) according to the previously mentioned method. A total of 500 µL of fungal suspension was incubated with GmAMP at concentrations (25, 50, and 100 µM) in 24 well plates and 10 mM PBS as the negative control. Cells were co-incubated with peptides for 3, 6, 9, 12, and 24 h at 37 °C, then the hypha formation was observed and photographed under an inverted microscope.

Antibiofilm assay

Fungal cells grown to the logarithmic phase were adjusted to 1.0 × 106 CFU/mL with RPMI-1640 liquid medium and added to 96-well polypropylene plates, which were then incubated at 37 °C for 90 min (biofilm formation assay) or 48 h (biofilm eradication assay) according to previous method (Zou et al., 2024). And 100 µL newly prepared 2, 3-bis(2-methoxy-4-nitro-5-sulfophenyl) 2H-tetrazolium5-carboxamide sodium salt (XTT) solution (Yuan Ye, Shanghai, China) was added to each well for 2 h at 37 °C after different concentrations of GmAMP were added and continued to co-incubate for 24 h. A microplate reader set to OD490nm was used to measure absorbance.

Sterile polylysine cell crawls were then placed on the bottom of a 24-well plate and 500 µL of the fungal suspension at the above concentration was added. The biofilm formation and mature biofilm were prepared according to the above method, and 500 µL of SYTO 9 (Invitrogen, Waltham, MA, USA) and propidium iodide (PI; Sigma) solution with the final concentration of 10 µM were added and incubated for 20 min while 10 mM PBS was used as the negative control. Seal the cover glass with nail polish, laser confocal microscopy (Olympus SpinSR10; Olympus, Tokyo, Japan) was used to observe and obtain images.

Scanning electron microscope

The concentration of the prepared fungal cells was 1.0 × 106 CFU/mL based on the previous description with minor modifications (Alfaro-Vargas et al., 2022), incubated in GmAMP with culture medium at 35 °C for 2 h, and 10 mM PBS was used as a negative control, then the suspension was centrifuged at 5,000 rpm for 10 min, fixed with 2.5% glutaraldehyde overnight at 4 °C, and dehydrated with 50%, 75%, 95% and 100% series of ethanol solutions for 10 min. It was then dried in a vacuum evaporator and coated with a thin layer of gold-palladium. The samples were observed by scanning electron microscopy and the image acquisition was performed using a Hitachi Regulus SU8100 (Tokyo, Japan).

Flow cytometry analysis

The concentration of the prepared fungal cells was 1.0 × 106 CFU/mL according to the previous description with minor modifications (Torres et al., 2023), the GmAMP with different concentrations was incubated at 35 °C for 1 h, and 10 mM PBS was used as the negative control. After that, the fungal suspension was incubated with SYTO 9 and PI staining with a final concentration of 10 µM at 35 °C for 15 min, and the stained cells were analyzed by flow cytometry.

Membrane potential

The concentration of the prepared fungal cells was 1.0 × 106 CFU/mL as described in the previous study (Decker et al., 2024), then added to 96-well plates while the membrane potential DiSC3(5) probe was added into the fungal suspension. The fungal suspension was then treated with different concentrations of GmAMP, and 10 mM PBS was used as the negative control for measured fluorescence intensity. The change of fluorescence intensity in 1 h was continuously and dynamically monitored with by RF-5301PC sectrofluoro-photometer (Bio-Tek Synergy HTX, Winooski, Vermont, USA).

Reactive oxygen species level

The levels of ROS were determined by using 2′, 7′-dichlorodihydrofluorescein diacetate (DCFH-DA; Yuan Ye, Shanghai, China) according to previously described methods (Shaban, Patel & Ahmad, 2024). The 1.0 × 106 CFU/mL of fungal suspension was added to 96-well plates while the Reactive Oxygen DCFH-DA probe (10 µM) was added into the fungal suspension. The fungal suspension was then treated with different concentrations of GmAMP. 10 mM PBS and 10 µM NAC (N-Acetylcysteine) were used as the negative and positive control. The change of fluorescence intensity in 1 h was continuously and dynamically monitored by RF-5301PC sectrofluoro-photometer (Bio-Tek Synergy HTX, Winooski, VT, USA).

Cytotoxicity Assays

Mouse RAW 264.7 cells were used to evaluate the cytotoxicity of GmAMP on mammalian cells as previously described (de Oliveira et al., 2023). Cells were cultured in Dulbecco’s Modified Eagle Medium (DMEM, Gibco, Grand Island, NY, USA) containing 10% fetal bovine serum (FBS; Gibco, Waltham, MA, USA), 1% penicillin-streptomycin (Gibco, Waltham, MA, USA), and maintained at 37 °C in a humidified 5% CO2 incubator. Firstly, 100 µL of RAW 264.7 cells suspension (2 × 104 cells/mL) was added to 96-well plates for cultivating overnight. A total of 100 µL with different concentrations of GmAMP solution was added and then incubated at 37 °C for 24 h. 10 mM PBS and complete medium were used as the negative and blank controls, respectively. When the incubation period was over, 10 µL of CCK8 solution was added to each assay well following the manufacturer’s protocols (MedChemExpress, Monmouth Junction, NJ, USA) and incubated for 1 h. Absorbance values were detected at OD450nm, and the percentage of cell survival was counted.

Cellviability(%)=(Abs450nmofGmAMPsolution−Abs450nmofblankcontrolAbs450nmofPBScontrol−Abs450nmofblankcontrol)×100%.

Hemolysis of human red blood cells

The human red blood cells (hRBCs) were used to evaluate the hemolytic activity of GmAMP based on previous descriptions with minor modifications (Larrán et al., 2022). The GmAMP with different concentrations and 2% hRBCs were added to the 96-well plate and incubated for 1 h in 37 °C. 1% Triton X-100 (Solarbio, Beijing, China) was the positive control, and 10 mM PBS was used as the negative control. When the incubation period was over, as the samples were gathered and centrifuged for 10 min at 1,000 rpm, the supernatant was moved to a fresh 96-well plate, and the OD540nm absorbance value was used to determine the extent of hemolysis. The hemolysis test of the new peptide has been approved by Ethics Committee of GuiZhou Medical University, approval number: 2022 Ethical Approval (302).

Hemolysis(%)=(Abs540nmofGmAMPsolution−Abs540nmofPBScontrolAbs540nmof(TritonX−100)−Abs540nmofPBScontrol)×100%.

Galleria mellonella infection model

The larvae used in the experiment were purchased from Huiyude Biotechnology Co., Ltd., (Tianjin, China), each weighing 250~300 mg and about 2~3 cm in length. As previously described with slight modifications (Fernandes, Weeks & Carter, 2020). All larvae were placed in a dark incubator at 35 °C overnight before the experiment. Ten larvae were randomly divided into each group to inject 10 µL of GmAMP solution with a concentration of 8~32 mg/kg into the last left proleg of larvae to evaluate the toxicity of GmAMP. The negative control was given an equal volume of sterile PBS. In order to assess the effectiveness of GmAMP, 12 larvae were randomly assigned to each group, and 10 µL of a fungal suspension containing roughly 5.0 × 108 CFU/mL was injected into the final left proleg of the larvae. After 1 h in the incubator, the same volume of GmAMP was injected into the last right proleg using the same method. Live and dead counts were taken every 24 h during the five days of incubation at 35 °C. Larvae were deemed dead when they went dark or mushy and had no discernible tactile reaction. Three larvae per group were chosen at random and placed into 1.5 mL of sterile PBS solution for high-speed homogenization and grinding after 24 h following GmAMP injection. The 10 µL gradient dilution was dropped onto a sterile solid YPD plate and incubated for 24 h. The number of fungal single colony was recorded, and the C. tropical burden of each larva was counted.

Data processing

Statistical mapping and data analysis were performed using GraphPad Prism 8.0 software (GraphPad Software). Data were expressed as mean ± SD and analyzed by one-way ANOVA. Long-rank test was used for the analysis of Mantel-Cox survival curves for the G. mellonella survival experiment. P < 0.05 was considered statistically significant.

Results

GmAMP chemical characteristics

The antibacterial activity of antimicrobial peptides is significantly influenced by their secondary structure. The antimicrobial peptide’s interaction with the cell membrane is facilitated by the alpha-helical structure, which increases the antibacterial activity. (Personne et al., 2023). The antimicrobial peptide GmAMP, composed of 21 amino acids, is predicted to have an α-helical structure (Figs. 1A and 1B). The molecular weight (MW) of GmAMP was confirmed to be 2,539.35 Da by mass spectrometry. Furthermore, GmAMP has a net charge of +17 and a hydrophobicity value of −0.757 (Fig. 1C), these characteristics imply that GmAMP is a cationic hydrophilic peptide.

Figure 1 Physicochemical properties of GmAMP.

(A) Helical wheel analysis of GmAMP. Positively charged amino acids are indicated in blue, the red ‘N’ represents the starting position and the arrow represents the hydrophobic moment. (B) Predicted three-dimensonal spatial structures of GmAMP. (C) The physical and chemical properties of GmAMP.

Antifungal activity

The results of the antifungal activity assay showed that GmAMP had strong antimicrobial effects against clinical fluconazole-resistant C. tropicalis, with MICs ranging from 25 to 50 µM (Table 1). In comparison to this, the MIC values of fluconazole against C. tropicalis ATCC 20962 was 13.06 µM (4 µg/mL), which was consistent with the Clinical and Laboratory Standards Institute standard (CLSI). The MIC values of fluconazole against clinical isolates exceeded 3,343 µM. Consequently, to gain a more comprehensive understanding of the impact of GmAMP on fluconazole-resistant Candida tropicalis clinical isolate, the 4252 isolate (MIC = 25 µM) was selected for further investigation.

Table 1 Antifungal activity of GmAMP.

Determination of antifungal activity of GmAMP against five strains of Candida tropicalis.

Strains	MIC (µM)	
GmAMP	Fluconazole	
Fluconazole-resistant C. tropicalis 4252	25	>3,343	
Fluconazole-resistant C. tropicalis 4171	50	>3,343	
Fluconazole-resistant C. tropicalis 6984	50	>3,343	
Fluconazole-resistant C. tropicalis 8402	50	>3,343	
C. tropicalis ATCC 20962	12	13	

Growth kinetics and fungicidal kinetics

To elucidate the effect of GmAMP on the growth process of fluconazole-resistant C. tropicalis, the growth curve of fluconazole-resistant C. tropicalis under GmAMP treatment was further plotted, as shown in Fig. 2A, the fungal cells in the control group entered the logarithmic phase within 6 h and reached the stationary phase within 18 h. In comparison, GmAMP treatments at concentrations of 25 and 50 µM were found to slow down the proliferation rate of fluconazole-resistant C. tropicalis and prolong the time to reach the logarithmic phase. When treated with GmAMP at a concentration of 100 µM, GmAMP showed an inhibitory effect on fluconazole-resistant C. tropicalis and prevented the natural growth and reproduction of fluconazole-resistant C. tropicalis. The time-fungicidal kinetic curve was further plotted to clarify the fungicidal effect of GmAMP (Fig. 2B). Compared with the control group, GmAMP exhibited a powerful inhibitory effect on fluconazole-resistant C. tropicalis when the concentrations of GmAMP were 25 and 50 µM. Notably, dealing with the concentration at 100 µM of GmAMP, the fluconazole-resistant C. tropicalis was killed within 2 h. These results indicate that GmAMP manifests effective antimicrobial activity and fungicidal effect against fluconazole-resistant C. tropicalis. Low-concentration GmAMP exerts an inhibitory effect on the growth of drug-resistant Candida tropicalis, whereas high-concentration high GmAMP directly kills Candida cells.

Figure 2 Effect of GmAMP on the growth of fluconazole-resistant C. tropicalis.

(A) Growth kinetics of fluconazole-resistant C. tropicalis. (B) Time-killing kinetics of fluconazole-resistant C. tropicalis. (C) The transformation from yeast phase to mycelial phase of fluconazole-resistant C. tropicalis. Scale bar, 25 µm.

The transformation of yeast to the mycelial phase

The transformation of Candida mycelial morphology is closely related to its pathogenicity. Morphological changes during the transformation of the fluconazole-resistant C. tropicalis yeast phase to the mycelial phase were observed by using an inverted microscope which is shown in Fig. 2C. The length of the mycelium of the fungus in the control group increased with incubation time, after 9 h of incubation, the yeast cells were observed to grow and form bundles of hyphae while forming branches of various sizes and lengths and intertwining with each other to form a net structure, and a more tightly netted biofilm is formed over time. Interestingly, the development of fluconazole-resistant Candida tropicalis into its hyphal form was inhibited to varying degrees following treatment with GmAMP. Specifically, the morphological transformation from yeast to hyphae was partially inhibited at concentrations of 25 and 50 μM of GmAMP, whereas it was completely suppressed at a concentration of 100 μM. These results suggest that GmAMP inhibited the morphological transformation process of fluconazole-resistant C. tropicalis from the yeast phase to the mycelial phase.

Inhibition of biofilm formation and eradication of mature biofilm

In order to investigate whether GmAMP has an anti-Candida tropicalis biofilm effect, we visualized the results by using confocal laser scanning microscopy. In the control group of the biofilm formation assay, a compact and intact biofilm was observed to emit predominantly green fluorescence colored by SYTO9, a dye that penetrates cells. Nevertheless, after GmAMP treatment, intact biofilm could no longer be formed and only dispersed incomplete membranes of different sizes were observed, while only single yeast cells could be seen in the high-concentration group (Fig. 3A). In the control group of the biofilm eradication assay, it was observed that the tightly structured and complete biofilm mainly emitted green fluorescence colored by SYTO9. After GmAMP treatment, we found that the tightly structured biofilm became loose and the network structure was reduced and thinned, while a reduction in the number of yeasts and an increase in the number of fungal damages was observed, with a predominantly red fluorescence colored by PI, a dye that penetrates injured cells (Fig. 3B). In addition, the biofilm activity of C. tropicalis was determined by XTT quantitative method. Compared with the control group, GmAMP at concentrations of 50, 100, and 200 µM inhibited biofilm formation by 49.75%, 71.28%, and 88.32%, respectively (Fig. 3C), and eradicated mature biofilm by 17.53%, 42.07%, and 58.28%, respectively (Fig. 3D). These results indicated that GmAMP inhibited biofilm formation and eradicated a certain amount of mature biofilm in fluconazole-resistant C. tropicalis.

Figure 3 Effect of GmAMP on the biofilm of fluconazole-resistant C. tropicalis.

Inhibition (A) and eradication (B) effects of fluconazole-resistant C. tropicalis biofilms treated with GmAMP at different concentrations observed by confocal laser scanning microscopy. Images obtained by live/dead staining (SYTO 9, green; PI, red). Scale bar, 20 µm. The activity level of biofilm under different concentrations of GmAMP was determined by the XTT reduction method (C and D), and the colorimetric absorbance was measured at OD490nm. The error bar represents the standard deviation of the three independent experiments. ***P < 0.001 compared with the control group.

Antifungal mechanism

To investigate the impact of GmAMP on the cell morphology of fluconazole-resistant C. tropicalis, the morphological changes induced by GmAMP treatment of fungal cells for 2 h were directly observed by scanning electron microscopy. Untreated fungal cells were morphologically intact, with cell surfaces remaining round and smooth (Fig. 4A). When cells were exposed to 100 µM of GmAMP for 2 h, the cells were destroyed and the surface appeared rough and irregular (Fig. 4B). Thus, these outcomes suggested that the cellular structural integrity of fluconazole-resistant C. tropicalis has been impaired. To further investigate the interaction of GmAMP on the cell membrane of fluconazole-resistant C. tropicalis, PI and SYTO9 fluorescence staining were used to determine the effect of GmAMP on the integrity of the cell membrane. PI and STOY9 are DNA-binding dyes emitting red and green fluorescence, respectively, and the former only penetrated the membrane-damaged cells, while the latter stained both live and dead cells (Jin et al., 2005). As a result, when cells were exposed to 25, 50, and 100 µM of GmAMP, 45.02%, 76.88%, and 85.02% of the cells stained positively for PI, respectively. Moreover, PI staining positivity showed a dose-dependent correlation with GmAMP (Fig. 4D), which indicated that GmAMP disrupted the cell membrane integrity of fluconazole-resistant C. tropicalis. The membrane depolarization was detected using the membrane potential probe DiSC3(5). As shown in Fig. 4E, compared with the control group, we found that GmAMP treatment caused depolarization of the cell membrane potential of fluconazole-resistant C. tropicalis and the level of membrane potential rose significantly with increasing concentration of GmAMP. Reactive oxygen species (ROS) generally maintain low levels within normal cells, but the accumulation of higher levels of ROS can damage cellular structures (Huang et al., 2020). Here, different concentrations of GmAMP induced ROS accumulation in fluconazole-resistant C. tropicalis, and ROS levels showed a time-dose dependence (Fig. 4F). These results suggest that the presence of GmAMP induced ROS production, which contributed to the crucial factor in the antimicrobial effect of GmAMP.

Figure 4 Effects of GmAMP cell morphology and cell membranes of fluconazole-resistant C. tropicalis.

The control group (A) and GmAMP group treated with 100 µM (B) of fluconazole-resistant C. tropicalis morphological images by scanning electron microscopy. (C) Cell membrane permeability of GmAMP on the fluconazole-resistant C. tropicalis was determined by flow cytometry, and with SYTO 9 and PI as pore formation mechanism marker. (D) The bar chart showed the percentage of PI positive cells. (E) DiSC3(5) was used to detect the cell membrane depolarization of fluconazole-resistant C. tropicalis. (F) The ROS-induced accumulation of DCFH-DA is a pore formation mechanism marker. The error bar represents the standard deviation of the three independent experiments. *** P < 0.001 compared with the control group.

Hemolytic and cytotoxicity

To evaluate mammalian cytotoxicity, we assessed the safety of GmAMP on human red blood cells and RAW 264.7 cells. The hemolysis experiment was performed using 2% human red blood cells. At a concentration of 200 µM, GmAMP exhibited slight hemolysis with a hemolysis rate of 35.64% (Fig. 5A). Moreover, GmAMP showed no obvious cytotoxicity to RAW 264.7 cells, and the cell viability of mouse macrophage RAW 264.7 remained above 80% at 200 µM concentration of GmAMP (Fig. 5B).

Figure 5 The hemolytsis and cytotoxicity effects of GmAMP.

(A) The cytotoxicity of GmAMP against RAW 264.7 cells. (B) The hemolysis rate of 2% human red blood cells.

Therapeutic effect on fluconazole-resistant C. tropical infection in vivo

The G. mellonella larvae infection model was used to investigate the treatment efficacy of GmAMP in vivo. In the peptide toxicity test (Fig. 6B), all larvae survived, suggesting that GmAMP did not exhibit significant toxicity within the 32 mg/kg dosage range. Survival of larvae infected with fluconazole-resistant C. tropicalis was increased by injecting various concentrations of GmAMP, with 75% survival in the 32 mg/kg group (P < 0.05), whereas the control group had 40% survival rate at 5 days post infection (Fig. 6C). These results were also reflected by the fungal burden of G. mellonella larvae. The number of colonies per larva was significantly reduced in all treatment groups after 24 h of treatment with GmAMP, whereas 32 mg/kg GmAMP reduced the fungal burden from 5.27 × 108 to 6.37 × 106 CFU per larva (Fig. 6D). Consequently, GmAMP has the potential for clinical application as it effectively treats fluconazole-resistant C. tropicalis infection and reduces the fungal burden in vivo.

Figure 6 In vivo toxicity and therapeutic activity of GmAMP in the G. mellonella model.

(A) Schematic diagram of the GmAMP treatment. (B) The toxicity of GmAMP in G. mellonella larvae model. (C) Survival of larvae after treatment with GmAMP. (D) Fungal burden of larvae after treatment with GmAMP. *P < 0.05; ***P < 0.001 compared with the group of fluconazole-resistant C. tropicalis + PBS.

Discussion

Antifungal drug resistance is currently rapidly developing due to the abuse of antibiotics, which is increasing the morbidity and fatality rate from invasive fungal infections (Fan et al., 2024). The isolation rate of Candida species, such as Candida albicans, Candida glabrata, Candida tropicalis, Candida krusei, and Candida parapsilosis is steadily increasing in hospitals (Falagas, Roussos & Vardakas, 2010; Lee et al., 2022). According to recent reports, China’s proportion of isolates of C. tropicalis that are resistant to fluconazole is still increasing (Wang et al., 2021). The development of new antifungal drugs to address this problem is imminent. Widely distributed in animals, plants, and other organisms, AMPs are a fast and effective barrier against pathogens in humans. AMPs exert their antimicrobial activity through a unique membrane-targeting mechanism that avoids the development of drug resistance and is regarded as a novel alternative to synthetic antibiotics (Mulukutla et al., 2024).

In this study, we determined in vitro the antimicrobial activity of GmAMP against four clinical isolates of fluconazole-resistant C. tropicalis and the standard strain of C. tropicalis ATCC 20962, and GmAMP showed good antimicrobial effect against clinical isolate of fluconazole-resistant C. tropicalis 4252, with a MIC value of 25 µM. GmAMP not only delayed the proliferation rate and inhibited the growth and reproduction of fungi, but also effectively killed C. tropicalis within 2 h. These results suggest that GmAMP is an effective antimicrobial agent and further studies on the antifungal efficacy of GmAMP are necessary.

The process of transformation from the yeast phase to the mycelial phase, termed “biphasic”, is considered the most important pathogenic characteristic of Candida, and is also recognized as a key stage in biofilm formation and maturation (Zhu et al., 2024). Furthermore, the hyphae formed during the morphological transformation can penetrate cells and invade the bloodstream, expressing a wide range of virulence factors, and they are regarded as a more virulent phenotype than yeast (Khamzeh et al., 2023). Herein, GmAMP inhibited the transition of yeast phase cells to mycelial phase morphology, preventing the process of mycelial development, which demonstrates that GmAMP plays a key role in inhibiting the formation and maturation of fluconazole-resistant C. tropicalis biofilm by preventing mycelial development. The yeast cells of C. tropicalis are characterized by a high capacity to form biofilms compared to other Candida species (Zuza-Alves, Silva-Rocha & Chaves, 2017), which may be related to an increased amount of biomass in the membranes and extracellular matrix, leading to a denser structure (Chandra & Mukherjee, 2015; Desai & Mitchell, 2015). Here, the 50 µM (2 × MIC) of GmAMP inhibits biofilm formation and has an eradicative effect on mature biofilms. Moreover, GmAMP inhibited and eliminated biofilms of fluconazole-resistant C. tropicalis in a concentration-dependent manner. Therefore, GmAMP showed good bioactivity in inhibiting morphological transformation and anti-biofilm processes.

It has been reported that most antimicrobial peptides exert their antimicrobial effects mainly by targeting cell membranes (Aguiar et al., 2020; Buda De Cesare et al., 2020; Hu et al., 2022). In the present study, the results of the scanning electron microscopy assay demonstrated that GmAMP disrupts the morphology and structure of the fluconazole-resistant C. tropicalis. A similar phenomenon was also found by Ma et al. (2020), Zhang et al. (2023). Moreover, we speculated the exact reason for the morphological damage and subsequent cell death is correlated with increased membrane permeability resulting from electrostatic interactions between the positively charged peptide GmAMP and the negatively charged components of the fungal cytoplasmic membrane (Boparai & Sharma, 2020; Jayasinghe, Whang & De Zoysa, 2023; Kodedová et al., 2019). The experimental result verified membrane permeability by a significant increase in the number of PI-stained positive cells of the fungi treated with GmAMP. Changes in cell membrane permeability usually trigger variation in cell membrane potential, which is closely related to cellular function (D’Auria et al., 2022). When membrane-modifying compounds (e.g., peptides) depolarize the membrane then the potential is lost. DiSC3(5) is released into the solution, causing fluorescence enhancement, which indicates that the cytoplasmic membrane is altered because of the cell membrane depolarization by the action of GmAMP in a concentration-dependent manner, which suggests that the dissipation of membrane potential might be involved in the formation of channels or pores, then allowed the passage of ions or macromolecules, to lead cytoplasmic membrane dysfunction (Bezerra et al., 2022; Venkatesh et al., 2017). It has been found that aerobic metabolism-generated ROS are usually present in cells that are in equilibrium with antioxidant enzymes, and excess ROS have certain deleterious effects on the basic structure of fungi, such as damage to nucleic acids, DNA, amino acid residues, and cell membranes (Perrone, Tan & Dawes, 2008). We found that GmAMP induced the accumulation of reactive oxygen species in a dose-dependent manner (Taveira et al., 2022). In brief, we hypothesized that the cationic peptide GmAMP can interact with certain negatively charged substance molecules on the cell membrane through electrostatic interactions, it leads to a series of consequences such as increased membrane permeability, altered depolarization of the membrane potential, structural loss of membrane integrity, accumulation of ROS, and further leakage of intracellular contents, which finally leads to cytoplasmic membrane dysfunction and cell death.

The excellent antimicrobial activity of antimicrobial peptides is usually associated with strong hemolytic activity and cytotoxicity, and assessment of the in vitro safety of AMP is paramount for further consideration as a potential clinical candidate (Zhang et al., 2024). In this study, our results showed that GmAMP showed little cytotoxicity and low hemolytic effect. Although some cytotoxicity and hemolytic activity were observed at higher concentrations, considering the MIC value of GmAMP was 25 µM (Table 1), which was much lower than its cytotoxicity concentration, GmAMP safety is also guaranteed under the premise of ensuring its activity. However, the potential toxicity of GmAMP to other mammalian cells remains to be studied.

The safe and effective dose range of GmAMP is confirmed by cytotoxicity and hemolytic tests, laying the foundation for its application in animal studies. In this study, the therapeutic efficacy of GmAMP was tested in vivo using the G. mellonella larvae infection model, which showed a significant improvement in survival. The fungal burden of G. mellonella larvae was considerably decreased in vivo with GmAMP therapy. These findings suggest that GmAMP may exhibit a strong safety and certain therapeutic potential.

Conclusions

This work describes the antifungal activity and mechanism of antimicrobial peptide GmAMP and therapeutic efficacy in vivo. GmAMP possesses potent antimicrobial activity, anti-biofilm formation, and eradication ability, and may play an antimicrobial role by disrupting the structure of fungal cytomembrane. Here, GmAMP displays low cytotoxicity and low hemolytic activity in vitro experiments. Furthermore, GmAMP exhibits a therapeutic effect against fluconazole-resistant C. tropical infection and reduces the number of fungi in vivo. These properties make GmAMP a potential treatment for fluconazole-resistant C. tropical infection, which is worthy of further optimization and development. Furthermore, GmAMP provides further opportunities for the safe and effective clinical application of antimicrobial peptides in the development of drug resistance.

Supplemental Information

Supplemental Information 1 Determination of MIC value.

Supplemental Information 2 The raw data for Figure 2.

Supplemental Information 3 The raw data for Figure 3.

Supplemental Information 4 The raw data for Figure 4.

Supplemental Information 5 The raw data for Figure 5.

Supplemental Information 6 The raw data for Figure 6.

Supplemental Information 7 Characterization of GmAMP.

(A) purification of GmAMP. (B) electrospray ionization (ESI) mass spectrometer of GmAMP.

Additional Information and Declarations

Competing Interests

The authors declare that they have no competing interests.

Author Contributions

Ruxia Cai conceived and designed the experiments, performed the experiments, analyzed the data, prepared figures and/or tables, authored or reviewed drafts of the article, and approved the final draft.

Na Zhao conceived and designed the experiments, performed the experiments, prepared figures and/or tables, authored or reviewed drafts of the article, and approved the final draft.

Chaoqin Sun performed the experiments, analyzed the data, prepared figures and/or tables, and approved the final draft.

Mingjiao Huang performed the experiments, prepared figures and/or tables, and approved the final draft.

Zhenlong Jiao analyzed the data, prepared figures and/or tables, and approved the final draft.

Jian Peng performed the experiments, prepared figures and/or tables, and approved the final draft.

Jin Zhang analyzed the data, authored or reviewed drafts of the article, and approved the final draft.

Guo Guo conceived and designed the experiments, authored or reviewed drafts of the article, and approved the final draft.

Data Availability

The following information was supplied regarding data availability:

The raw measurements are available in the Supplemental Files.

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
