# Peer review of "Antifungal activity and mechanism of novel peptide Glycine max antimicrobial peptide (GmAMP) against fluconazole-resistant Candida tropicalis"

_PeerJ, doi:10.7717/peerj.19372_

## Round 0.1 · original submission · Major Revisions

The two expert reviewers did an excellent job of critiquing your work with good suggestions. Overall, they felt the work was solid but the structure of the paper, formatting, citations and grammar need to be substantially improved.

·

Basic reporting

1. All in-text citation: please follow ‘Name, year’ format, e.g. (Smith et al., 2005) with name and year separated by coma.
2. Multiple referenced sources were not consistent with the arguments that authors were trying to make. Several references used in the method section didn’t contain the referred method. Please check all references for accuracy.
3. The links included in reference section don’t work. Please refer to the peerJ guidance for reference format https://peerj.com/about/author-instructions/#reference-format

Experimental design

1. What is the motivation to solely focus on C.tropicalis? It is clear in the background that C.tropicalis has a profound antifungal resistance issue, yet other candidate species also display a trend in increased antifungal resistance. Have the authors tested other Candida species? Is there any reason to believe the peptide is a narrow spectrum antifungal?
2. Methods of antifungal assay lack some important information for accurate interpretation – please refer to pdf for details

Validity of the findings

1. Some discrepancies between initial MIC values from microbroth dilution assay and the further characterizations were found and should be addressed.
2. Interpretation of data is not accurate, comparing figure 2A/2B to the result text section – see annotation in pdf.
3. Some inconsistent languages were used in describing assays and results, which added confusion. For example, in the biofilm assay section, an “inhibition” and “eradication” effects were mentioned, which may be from characterization at early and later stage of biofilm growth but wasn’t clear. Please clarify inconsistent language.

Reviewer 2 ·

Basic reporting

In the manuscript “Antifungal activity and mechanism of novel peptide GmAMP
against fluconazole-resistant Candida tropicalis” by Cai et al., the authors elegantly show the efficacy of GmAMP as a novel antifungal against Candida tropicalis. This manuscript is a continuation of a previous report identifying GmAMP as a potential therapeutic and does a very nice job of characterizing the antifungal activities of this peptide against fluconazole-resistant clinical isolates. The authors find that GmAMP has a dose-dependent impact on fungal killing and biofilm disruption, while showing minimal cytotoxicity in vivo. Collectively, this work expands on their previous findings identifying GmAMP as a potential novel therapeutic and does a very nice job laying the groundwork and rationale for why this peptide should be further studied moving forward.


Overall, the experiments are well thought through and robustly performed. However, certain formatting and experimental concerns need to be addressed and caveats of some of the conclusions drawn in the manuscript should be discussed.

Basic Reporting:

The authors do a very nice job throughout the manuscript in characterizing GmAMP as a novel peptide for C. tropicalis treatment. The overall structure of the paper flows logically and provides a compelling argument for GmAMP as a novel candidate for future validation. However, there are instances throughout the manuscript where grammar and sentence structure issues distract from the data being presented. I would recommend getting professional help to address these. Some examples include:
o Lines 35-36 needs a transition.
o Line 52: Candidiasis should not be italicized.
o Line 67: "Therapuetic" should be plural
o Line 88: In earlier period research work is a bit awkward, possibly change to “We have previously reported”?
o Line 94: Wet experiments should be wet lab experiments.
o Line 99: “The” should be lowercase.
o Line 132: a connecting word like “and” is needed after the comma.
o Line 137: “Resuspension” should be resuspended
o Line 199-200: “to be measured fluorescence intensity” reads improperly.
o Line 204: “according to previously described with..” should be "according to previously described methods"
o Lines 274-276: Sentence delineation here is a bit sporadic, and periods are introduced seemingly in the middle of a sentence.
o Line 355: there should be no period after “(Fig 6B)”
o Line 361: “number of colonies pre larvae” should be "per larvae"
o Lines 382-383: The text reads “but also effectively killed GmAMP within 2 h”. Do the authors mean GmAMP or C. tropicalis?

The authors do a very nice job citing literature and supporting their rationale throughout the introduction of the paper. However, there are several sections that would be strengthened by statements that further expand on the rationale for the study. These include:
o Lines 85-88: It is unclear why the accepted mechanism for AMPs makes them less susceptible to resistance development. It may be beneficial to describe the mechanism here to bridge these two statements.
o In the introduction, it is a bit unclear what information was reported in the last publication and where the new information in this manuscript begins. It would be beneficial to delineate these by adding a statement (such as “in this report”) where the work from the current study begins.

When discussing growth kinetics of fungi, various terms are used to describe the later stages of growth where cells stop replicating exponentially. In the paper this is referred to as the “lethargic phase” or the “stabilization phase”. Traditionally, this phase is referred to as the stationary phase of growth for cells. The authors should use uniform nomenclature when referring to this stage of growth throughout the entirety of the manuscript.

The image quality in Figure 2C is quite low and make it difficult to thoroughly visualize differences in fungal morphology. The resolution of these images should be increased.

Experimental design

The authors do a very robust job in both designing and executing experiments to further assess the impact that GmAMP has on C. tropicalis. However, there are several areas where more information is needed.

o In Figure 1, the programs used to generate the figures are listed in the figure legends. However, this is the not the appropriate place to provide that information, and the methods used to generate these predictions should be clearly stated in the methods section of the manuscript. Additionally, more detail is need in the methods for how mass spectrometry was used to determine the molecular weight of the compound.

o Figure 4C shows a nice trend between SYTO9 and PI staining with increasing drug concentrations. However, the authors need to explain how they gated their samples following flow analysis. Often, cell stress and damage can cause fungal cells to form aggregates, and if size is not accounted for in the gating strategy than the increased red signal may be simply the consequence of increased aggregation of cells giving off a higher emission for PI staining.

o Line 305: The “inhibition of biofilm formation and eradication of mature biofilm” section would be strengthened with a sentence stating the rationale for the experiments.

Validity of the findings

Overall, the authors do an excellent job interpreting their data and drawing conclusions from their results. Yet, there are several issues that should be addressed moving forward:

In line 268 the authors reference the results from the antifungal activity assay. However, it is unclear if this result was from the current study, or the previous one identifying the GmAMP compound. If the former, the authers should reference the table associated with this finding. If the later, this needs to be clearly stated in the text that was previously observed.

In figure 2C the authors nicely show the inhibitory effects of GmAMP treatment on cell morphology over time. They conclude that at higher concentrations (100uM) the drug reduced the number of yeast cells present (line 300). However, it is difficult to make this conclusion without showing a 0-hour timepoint as well. It is possible that it did not necessarily reduce the number of yeast cells, but simply prevented them from replicating (i.e. the number of yeast cells is the same as the starting number in the experiment). The image quality is also quite low in the figure, but it appears that there are chains of yeast cells in the intermediate concentrations. It is difficult to tell due to image resolution, but could they possibly be pseudohyphae instead of yeast, and rather than a reduction in cell number they are observing yet another change in morphology at the intermediate concentrations?

In the “Inhibition of biofilm formation and eradication of mature biofilm” section of the manuscript the authors do a nice job showing the impact that GmAMP treatment has on biofilm formation and cell viability. However, the text body would be strengthened by explaining more of what the cell colors are indicative of. For example, the authors state “In the control group of the biofilm formation assay, a compact and intact biofilm was observed to emit predominantly green fluorescence.” What is the green a marker of and why is this indicative of a stable biofilm? Additionally, although there is clearly an increase in PI staining with drug treatment, PI is not necessarily a definitive marker for cell death, as cell stress can cause increased PI staining without actual death as well (PMID: 21199254).

In line 343-345 the authors state that “These results suggest that the presence of GmAMP induced ROS production, which contributed to the crucial factor in the antimicrobial effect of GmAMP.” While ROS levels are increased, the authors cannot definitively claim that this contributes to the mechanisms of GmAMP inhibition. To address this, the authors should pharmacologically inhibit ROS production and see if this in turn reduces the killing capacity of GmAMP.

Figure 5 is missing a legend to indicate what the red and green lines represent.

Annotated reviews are not available for download in order to protect the identity of reviewers who chose to remain anonymous.

---

## Round 0.2 · Minor Revisions

HI, if you can just add in methods for the antioxidant (NAC) experiment you did, we will be set to go. I do not have to send out to reveiwers again.

·

Basic reporting

The authors revised in accurate citations.

Experimental design

Methods were revised to add more details on experimental set-up.

Validity of the findings

1. The interpretation of data shown in 2A/2B still didn’t accurately represent the graphs – see previous comments in pdf (line 307-310).
2. Inconsistent language was carefully addressed – the text was clearer now.

Additional comments

The authors carefully revised the manuscript and address major language usage and inaccurate interpretation. I suggest accepting the manuscript after minor edits (see “validity of the findings” point 1).

Reviewer 2 ·

Basic reporting

The authors have done an excellent job addressing the reviewers comments for this section. I have no additional major suggestions, but minor observations that should be edited before publication.

Lines 64 and 408: C. tropicalis needs to be italicized.

Line 391: “In the peptide toxicity test (Fig. 6B), in the experimental model,” These two statements are redundant, and only one is needed.

Experimental design

The authors have done and an excellent job and addressed all the reviewer comments adequately. However, information on NAC treatment (which is data added to the revised manuscript) needs to be added into the methods section for ROS levels.

Also, in line 187 it reads “Sterile polylysine cell crawls were then placed on the bottom of a 24-well plate and 500 µL of the fungal suspension at the above concentration was added.” Is this the start of a new experimental method for they Syto9 and PI staining or a follow up to the fluorescence biofilm read? If it is the start of another assay then it should start a new paragraph.

Validity of the findings

The authors do well interpreting their data and have addressed all the reviewer comments.

---

## Round 0.3 · accepted · Accept

I am satisfied with the current version.